# CAR-T Cell Therapy for the Treatment of ALL: Eradication Conditions and In Silico Experimentation

**Paul A. Valle** [1,*] , **Luis N. Coria** [1] , **Corina Plata** [1] **and Yolocuauhtli Salazar** [2]

1   Postgraduate Program in Engineering Sciences, BioMath Research Group, Tecnológico Nacional de México/IT Tijuana, Blvd. Alberto Limón Padilla s/n, Tijuana 22454, Mexico; luis.coria@tectijuana.edu.mx (L.N.C.); corina.plata@tectijuana.edu.mx (C.P.)
2   Postgraduate Program in Engineering, Tecnológico Nacional de México/IT Durango, Blvd. Felipe Pescador 1830 ote., Durango 34080, Mexico; ysalazar@itdurango.edu.mx
*   Correspondence: paul.valle@tectijuana.edu.mx

**Abstract:** In this paper, we explore the application of Chimeric Antigen Receptor (CAR) T cell therapy for the treatment of Acute Lymphocytic Leukaemia (ALL) by means of in silico experimentation, mathematical modelling through first-order Ordinary Differential Equations and nonlinear systems theory. By combining the latter with systems biology on cancer evolution we were able to establish a sufficient condition on the therapy dose to ensure complete response. The latter is illustrated across multiple numerical simulations when comparing three mathematically formulated administration protocols with one of a phase 1 dose-escalation trial on CAR-T cells for the treatment of ALL on children and young adults. Therefore, both our analytical and in silico results are consistent with real-life scenarios. Finally, our research indicates that tumour cells growth rate and the killing efficacy of the therapy are key factors in the designing of personalised strategies for cancer treatment.

**Keywords:** acute lymphocytic leukaemia; cancer eradication conditions; CAR-T cell therapy; complete response; in silico; mathematical modelling; personalised treatment



## 1. Introduction

Acute leukaemias are classified into myeloid and lymphocytic, depending on the result of the immunophenotype characterisation. Concerning acute lymphocytic leukaemias (ALLs), the latest World Health Organization classification replaced the classic cytological classification into B cell ALL and T cell ALL. Although, both are a malignant transformation and proliferation of lymphoid progenitor cells (B or T lymphoblasts) affecting mainly the bone marrow and peripheral blood. ALL is thought to be the result of a damage to DNA that causes lymphoid cells to undergo a rapidly and uncontrolled growth replacing elements from the bone marrow and other lymphoid organs, ultimately spreading throughout the body [1,2]. Furthermore, leukaemia cells do not have the ability to fight pathogens as well as a normal white blood cell would and possess an increased maximum lifespan, which is one of the main characteristics of a cancer cell.

Although ALL affects mostly children with 80% of all registered cases, the prognosis in paediatric patients is excellent with an overall 5-year survival rate of 69.9% and 90–95% of cases achieving complete response (CR) after chemotherapy. Nonetheless, it represents a devastating disease when it occurs in older adults, where the highest percentages of deaths occur after 65 years of age [2–5].

Concerning the treatment of ALL [4,6–8], incremental advances in leukaemia therapies have led to marked improvements in survival, abolishing the prognostic influence of clinical and biologic variables that were previously related to a poor outcome. Currently available treatments for ALL include, mainly, chemotherapy and stem cell transplantation. In the particular case of chemotherapy, it consists of induction, consolidation and long-term maintenance. The goal of induction therapy is to achieve CR and restore normal

haematopoiesis. Consolidation therapy aims to eradicate the submicroscopic residual disease that remains after CR. Maintenance therapy is the final stage of treatment, this phase has been demonstrated to lower the risk of relapse once CR has been established. However, 15% of children do not respond well to this approach [9], and untreatable relapse remains as a leading cause of cancer deaths in this age group. In adult patients, relapse is more frequent and often unsalvageable. Elderly patients are often unable to tolerate such regimens and carry a particularly poor prognosis. Relapsing patients usually have had already the maximum tolerable dose of chemotherapy. Thus, there is a pressing need for novel strategies in the overall treatment and relapse in ALL.

In recent years, chimeric antigen receptor (CAR) T cells therapy has revolutionised the treatment of resistant haematological malignancies and quickly became a new standard of treatment for relapsed/refractory disease [10]. This therapy has been investigated in both solid and non-solid tumours with promising results due to many studies showing high rates of remission [11–15]. Furthermore, CAR-T cell technology has the potential to be applied for the treatment of numerous diseases such as hemophilia, type 1 diabetes, multiple sclerosis, influenza A, HIV, SARS-CoV-2 and cardiac fibrosis, among others [16]. At a cellular level, CAR-T cells are generated by the T cells from either a patient or a donor's blood. After these cells are expanded and genetically modified they are reinfused into the cancer patients to specifically target and destroy tumour cells [17]. CARs are engineered receptors that can graft an arbitrary specificity onto an immune effector cell [17,18]. Unfortunately, CAR-T cells are associated with serious toxicity that includes cerebral oedema, cytokine release syndrome, on target/off tumour recognition, neurological toxicity, cytopenia and anaphylaxis [1,19–21]. Thus, there are still some challenges that need to be resolved concerning maximum tolerated dose, schedules of application, toxicity management and long-term effects of this treatment.

On the last subject, mathematical modelling can provide a powerful tool to further investigate both short- and long-term effects of CAR-T cells therapy administration. Additionally, mathematical models could be used to solve both the dosing problem and the scheduling for the intervals of application. However, for these models to be applicable in real-life scenarios they need to be formulated from experimental data and/or clinical trials. In the particular case of leukaemia and its treatment, Chulián et al. performed an extensive review concerning several mathematical models of first-order Ordinary Differential Equations (EDOs) in the literature [22]. Among these models, we found two of particular interest where CAR-T cells therapy is applied for the treatment of B cell ALL [23] and T cell ALL [24]. From these models, we formulated two first-order ODEs to explore the periodical applications of CAR-T cells and the corresponding effect on the time-evolution of ALL. In order to validate this approach, we compared our results with those reported by Lee et al. [11] in their clinical trial on CAR-T cells therapy for the treatment of ALL in children and young adults.

In the scope of our system, we were able to solve the dosing problem on the CAR-T cells therapy administration. This was achieved by applying nonlinear systems theories such as the Localization of Compact Invariant Sets (LCIS) [25,26] and Lyapunov's Direct [27,28] methods to establish a sufficient condition on the therapy dose, i.e., the minimum dose required to ensure both CR and complete eradication in the mathematical model under study in this work. Particularly, we designed three personalised therapy administration protocols for the application of CAR-T cells on a hypothetical patient with a given set of characteristics describing its overall health. The latter is illustrated by means of in silico experimentation [29–31], which is the process by which multiple numerical simulations are performed to compare different scenarios regarding initial tumour burdens, parameter values, total doses and intervals of therapy applications.

The remainder of this paper proceeds as follows. In Section 2, the necessary background on nonlinear systems theories to determine our mathematical results are presented. In Section 3, the ALL and CAR-T cells mathematical model is thoroughly described. In Section 4, bounds of the localizing domain are computed and sufficient conditions are

established on the CAR-T cells therapy dose to ensure nonexistence and eradication of the ALL cancer cells population. In Section 5, three therapy administration protocols are compared and discussed with one of a clinical trial, in silico experimentations illustrate that our mathematical results are consistent with real-life scenarios of cancer treatment. General conclusions of this work are given in Section 6, and we elaborate an Appendix A section to further explore the dynamical properties of the ALL and CAR-T cells system.

## 2. Materials and Methods

In this section, we provide the necessary background on nonlinear systems that allows us to derive sufficient conditions to ensure cancer cells eradication in this work. Further, when nonlinear systems theory is combined with systems biology, one can assume that there is a final critical value below which a population of cells cannot longer be considered as biologically meaningful. Therefore, results from previous works have enable us to formulate the next assumption on cell–cell interaction mathematical models.

**Assumption 1.** *Cells Eradication Threshold. See Section 4.2 in [31]. If a solution describing the growth of a cell population goes below the value of 1 cell, then it is possible to assume the complete eradication of such population.*

Assumption 1 is extremely relevant on the in silico experimentation phase as it is applied to determine the instant on which the cancer cells population is completely eradicated when performing numerical simulations. Therefore, the estimated time provides a threshold to stop the therapy administration protocol.

### 2.1. Localization of Compact Invariant Sets Method

Krishchenko and Starkov proposed the LCIS method in [25,26] to study the short- and long-time dynamics of nonlinear systems of first-order ODEs by computing the so-called localizing domain, which is a bounded region in the state space $\mathbf{R}^n$ where all compact invariant sets of a system are located.

The method is formulated as follows. Let us take an autonomous nonlinear ODEs system of the form $\dot{x} = f(x)$, where $f(x)$ is a $C^\infty$-differentiable vector function and $x \in \mathbf{R}^n$ is the state vector. Let $h(x) : \mathbf{R}^n \to \mathbf{R}$ be a $C^\infty$-differentiable function, $h|_S$ denotes the restriction of $h(x)$ on a set $S \subset \mathbf{R}^n$. The function $h(x)$ used in this statement is called localizing and it is assumed that $h(x)$ is not the first integral of $f(x)$. $S(h)$ denotes the set $\left\{ x \in \mathbf{R}^n \mid L_f h(x) = 0 \right\}$, where $L_f h(x)$ represents the Lie derivative (temporal derivative) of $f(x)$ and is given by $L_f h(x) = (\partial h/\partial x) f(x)$. From the latter, one can define $h_{\inf} = \inf\{h(x) \mid x \in S(h)\}$ and $h_{\sup} = \sup\{h(x) \mid x \in S(h)\}$. Hence, the General Theorem concerning the localization of all compact invariant sets of a dynamical system establishes the following.

**Theorem 1.** *General Theorem. See Section 2 in [26]. Each compact invariant set $\Gamma$ of $\dot{x} = f(x)$ is contained in the localizing domain:*

$$K(h) = \left\{ h_{\inf} \leq h(x) \leq h_{\sup} \right\}.$$

If the location of all compact invariant sets is inside the domain $\Lambda \subset \mathbf{R}^n$, then the set $K(h) \cap \Lambda$ may be formulated. It is evident that if all compact invariant sets are located in the sets $K(h_i)$ and $K(h_j)$, with $K(h_i), K(h_j) \subset \mathbf{R}^n$, then they are located in the set $K(h_i) \cap K(h_j)$ as well. Therefore, a refinement of Theorem 1 is realised with help of the Iterative Theorem stated as follows.

**Theorem 2.** *Iterative Theorem. See Section 2 in [26]. Let $h_m(x), m = 0, 1, 2, \ldots$ be a sequence of $C^\infty$—differentiable functions. Sets*

$$K_0 = K(h_0), \quad K_m = K_{m-1} \cap K_{m-1,m}, \quad m > 0,$$

*with*

$$K_{m-1,m} = \left\{ x : h_{m,\text{inf}} \leq h_m(x) \leq h_{m,\text{sup}} \right\},$$
$$h_{m,\text{sup}} = \sup_{S(h_m) \cap K_{m-1}} h_m(x),$$
$$h_{m,\text{inf}} = \inf_{S(h_m) \cap K_{m-1}} h_m(x),$$

*contain any compact invariant set of the system $\dot{x} = f(x)$ and*

$$K_0 \supseteq K_1 \supseteq \cdots \supseteq K_m \supseteq \ldots .$$

Nonetheless, if one considers the location of all compact invariant sets inside the domain $\Lambda \subset \mathbf{R}^n$ and $\Lambda \cap K(h) = \varnothing$, then it is possible to formulate the following Nonexistence Proposition.

**Proposition 1.** *Nonexistence Proposition. See Section 2 in [26]. If $\Lambda \cap K(h) = \varnothing$, then the system $\dot{x} = f(x)$ has no compact invariant sets located in $\Lambda$.*

Equilibrium points, periodic, homoclinic and heteroclinic orbits, limit cycles and chaotic attractors are examples of compact invariant sets. Localizing functions are selected by a heuristic process, this means that one may need to analyse several functions in order to find a proper set that will allow fulfilling Theorem 1.

### 2.2. Lyapunov's Direct Method

Aleksandr Lyapunov [27,28] concluded that a certain type of functions can be analysed to determine stability of an equilibrium point which is denoted as $x^* \in \mathbf{R}^n$ and satisfies $f(x^*) = 0$. Stability in the Lyapunov sense has a central role in the study of autonomous nonlinear ODEs systems of the form $\dot{x} = f(x)$, with $x \in \mathbf{R}^n$. Lyapunov's stability theorems provide sufficient conditions for stability and asymptotic stability of the equilibrium, both local and global.

In order to apply Lyapunov's direct method, it is necessary to formulate the so-called Lyapunov candidate function, which is usually denoted as $V(x) : \mathbf{R}^n \to \mathbf{R}$, a continuously differentiable function whose temporal derivative is given by $\dot{V}(x) = (\partial V / \partial x) f(x)$. This function must be positive definite, i.e., $V(0) = 0$ and $V(x) > 0$ for $x \neq 0$, while a negative definite function is also $V(0) = 0$, but $V(x) < 0$ for $x \neq 0$. Further, function $V(x)$ is said to be radially unbounded if $V(x) \to \infty$ as $\|x\| \to \infty$.

Now, by considering the information shown above, let us present the following theorem:

**Theorem 3.** *Global Asymptotic Stability. See Chapter 4 in [27] and Chapter 2 in [28]. The equilibrium point $x^*$ is globally asymptotically stable if there exists a function $V(x)$ positive definite, radially unbounded and decrescent such that its temporal derivative $\dot{V}(x)$ is negative definite.*

A function $V(x)$ satisfying the properties of Theorem 3 is called Lyapunov function. Nonetheless, there is no explicit method to find a candidate. Therefore, they should be formulated by trial and error. It is also important to mention that the fact that a Lyapunov candidate function fails to satisfy the requirements of Theorem 3 does not imply that the equilibrium must be necessarily unstable, it only means that stability in the sense of Lyapunov cannot be established.

### 2.3. Positiveness of Solutions

A dynamical system is said to be positive $[P]$ if and only if the non-negative orthant $\mathbf{R}^n_{+,0}$ is forward invariant, that is, for any non-negative initial conditions all trajectories remain either inside or at the boundaries of $\mathbf{R}^n_{+,0}$ for all future times [32,33]. Given an autonomous nonlinear ODEs system of the form $\dot{x} = f(x)$, the following Lemma provides a sufficient and necessary condition to establish its positivity.

**Lemma 1.** *Positiveness of solutions. See Section II.A in [32]. The autonomous nonlinear system $\dot{x} = f(x)$ is positive if and only if the next holds*

$$P \; \forall \; x \in \partial \mathbf{R}_{+,0}^{n} \mid x = 0 \Rightarrow f(x) \geq 0.$$

The latter implies that when evaluating the vector function $f(x)$ at the boundary $\partial$ of $\mathbf{R}_{+,0}^{n}$, i.e., $f(x)|_{x=0}$, the result must be a non-negative function $[f(x) \geq 0]$. Therefore, given non-negative initial conditions, all solutions will have non-negative real values for all $t \geq 0$.

## 3. The ALL and CAR-T Cells Mathematical Model

Pérez et al. [24] and León et al. [23] formulated two mathematical models concerning CAR-T cell therapy for the treatment of T-ALL and B-ALL, respectively. These works explored the relation of these two cells populations interactions by means of nonlinear system theory in the form of first-order ODEs. The mathematical properties of each model were thoroughly analysed, including equilibrium points, local stability, the positiveness of solutions and the existence of periodic orbits. Further, parameter values and initial therapy dose incidence on the final dynamics of each system are illustrated and discussed.

The novelty of these systems relies mainly on the modelling of the CAR-T cells evolution. Authors included the following complex phenomena for this population: serial killing of leukaemia cancer cells due to direct encounters; fratricide, mutual killing between the cells preventing generation, expansion, and persistence; clonal expansion due to mitosis stimulation after recognising the target antigen, located in both leukaemia and other CAR-T cells; and a finite lifespan of between two weeks to one month. Additionally, note that the following two assumptions were made: CAR-T cells do not die after killing target cells and they do not expand in vivo. Expansion may be considered in the in vitro phase as cytokines are added externally forcing them to divide. Nonetheless, this process must be modelled separately under different conditions and assumptions.

The ALL cancer cells growth is modelled by means of the exponential law, which is not biologically applicable in the long-term. Therefore, in our approach, we apply a sigmoidal law as this provides an initial exponential growth phase with an upper bound given by the so-called maximum tumour carrying capacity [34]. This implies that a new parameter should be estimated. Concerning the CAR-T cells therapy, only one dose is applied as the initial condition. The latter leads to the conclusion that the initial dose of the therapy does not affect the final outcome on both CAR-T cells and leukaemia populations [23,24]. Therefore, we introduce a protocol administration parameter to consider further applications of the therapy in order to achieve complete eradication of the ALL population on the system. The ALL and CAR-T cells mathematical model is given by the next two first-order ODEs:

$$\dot{C} = \rho_C(L + C)C - \tau_C C - \alpha C^2 + \varphi_C, \tag{1}$$

$$\dot{L} = \rho_L L(1 - \tau_L L) - \alpha L C. \tag{2}$$

The dynamic of the CAR-T cells is formulated by Equation (1), and it is described as follows. The first term represents the stimulation to mitosis with a rate $\rho_C$ due to encounters with the target antigen in both ALL cancer cells and other CAR-T cells. The natural death rate of these cells is given on the second term by parameter $\tau_C$. The $\alpha$ parameter rate represents both fratricide, i.e., third term of Equation (1), and killing efficacy of leukaemia cells from the therapy, i.e., the second term in Equation (2). The value of this parameter is going to be estimated in the Discussion Section from a clinical trial and compared with others available in the literature. Further, of note is the fact that

$$\alpha > \rho_C, \tag{3}$$

providing a mathematical restriction on the overall proliferation of this cell population. Below, in Table 1, one can see that (3) always fulfils. The therapy administration protocol is given in the fourth term of Equation (1) by parameter $\varphi_C$. Sufficient conditions on the minimum dose will be derived by means of the LCIS and Lyapunov's Direct method. Now, as discussed above, the first term of Equation (2) represents the logistic growth of the ALL cancer cells population with a growth rate $\rho_L$ and the inverse of the maximum tumour carrying capacity given by $\tau_L$. Units, values and their ranges are shown in Table 1. Further, specific values for each parameter will be thoroughly discussed and justified at Section 5 by taking into account a clinical trial in order to perform the in silico experimentation by means of multiple numerical simulations. The latter is necessary to illustrate our mathematical results and compare them with real-life scenarios.

**Table 1.** Parameter information for the ALL and CAR-T cell therapy mathematical model.

| Parameter | Description | Values | Units |
|---|---|---|---|
| $\rho_C$ | Mitosis stimulation rate of CAR-T cells due to encounters with leukaemia cancer cells | $0 < \rho_C < \alpha$ | $(cells \times day)^{-1}$ |
| $\tau_C$ | Natural death rate of CAR-T cells | $\dfrac{1}{30} \leq \tau_C \leq \dfrac{1}{14}$ | $day^{-1}$ |
| $\alpha$ | Killing efficacy rate of CAR-T cells | $\alpha > 0$ | $(cells \times day)^{-1}$ |
| $\rho_L$ | Leukaemia cancer cells growth rate | $\dfrac{1}{60} \leq \rho_L \leq \dfrac{1}{20}$ | $day^{-1}$ |
| $\tau_L$ | Inverse of the maximum tumour carrying capacity | $\dfrac{1}{7.2 \times 10^{11}} \leq \tau_L \leq \dfrac{1}{4.64 \times 10^{11}}$ | $cells^{-1}$ |
| $\varphi_C$ | CAR-T cell therapy | $\varphi_C \geq 0$ | $cells \times day^{-1}$ |

The range on the maximum tumour carrying capacity $\left[\tau_L^{-1}\right]$ is estimated as follows. The human body contains approximately $(3.7 \pm 0.8) \times 10^{13}$ cells [35,36], where approximately 1.6% represents the population of lymphocytes, see Figure 6.1 in [35]. Thus, by direct proportion, one can compute $(5.92 \pm 1.28) \times 10^{11}$ total lymphocytes, and we assume that if the population of leukaemia cells reaches a value within this range, then it will inevitably imply the death of the patient as all healthy lymphocytes would have been replaced by malignant cells.

Now, note that by conditions of Lemma 1, the ALL and CAR-T cells system (1) and (2) is positive for any non-negative initial conditions. Therefore, any semi-trajectory is going to be positively forward invariant in the non-negative quadrant $\mathbf{R}^2_{+,0}$, and all dynamics are located in the following domain:

$$\mathbf{R}^2_{+,0} = \{C(t), L(t) \geq 0\}.$$

Furthermore, by Assumption 1, we have the following,

$$C(t) = 0 \; \forall \; C(t) < 1,$$

and

$$L(t) = 0 \; \forall \; L(t) < 1,$$

implying that if any solution goes below the value of 1 cell, then it is possible to consider the complete eradication of that population. Now, let us determine the tumour-free equilibrium point of Equations (1) and (2) by equating them to zero and solving for the state variable $C(t)$ when $L(t) = 0$. The result is shown below,

$$(C_0^*, L_0^*) = \left( \frac{\sqrt{\tau_C^2 + 4\varphi_C(\alpha - \rho_C)} - \tau_C}{2(\alpha - \rho_C)}, 0 \right), \tag{4}$$

and, as condition (3) always holds, it is evident that the tumour-free equilibrium (4) is non-negative $[C_0^* \geq 0, L_0^* = 0]$. In Appendix A, all remaining equilibria of the ALL and CAR-T cells system are calculated, i.e., another three equilibriums; conditions for persistence of the ALL cancer population are derived, i.e., necessary conditions to ensure that $L(t) > 0$; and existence and uniqueness of solutions for any biologically feasible initial conditions are established, i.e., $C(0), L(0) \geq 0$.

## 4. Results

### 4.1. Localizing Domain and Nonexistence Conditions

In this section, the lower and upper bounds for both CAR-T cells and leukaemia cells populations are formulated by means of three localizing functions. These bounds are given by inequalities in terms of the system parameters. Further, a nonexistence condition of compact invariant sets outside the plane $L = 0$ (tumour-free equilibrium point) is derived from the upper bound of the ALL cancer cells $[L(t)]$. This condition is established on the immunotherapy treatment dose $[\varphi_C]$.

Now, let us explore the first localizing function:

$$h_1 = C,$$

and compute its Lie derivative as indicated below,

$$L_f h_1 = \rho_C (L + C)C - \tau_C C - \alpha C^2 + \varphi_C,$$

from the latter, set $S(h_1) = \left\{ L_f h_1 = 0 \right\}$ may be written as follows by completing the square

$$S(h_1) = \left\{ \alpha \left( C + \frac{\tau_C}{2\alpha} \right)^2 - \frac{\tau_C^2}{4\alpha} = \varphi_C + \rho_C(L + C)C \right\},$$

now, it possible to estimate a lower bound by rewriting $S(h_1)$

$$S(h_1) \subset \left\{ \alpha \left( C + \frac{\tau_C}{2\alpha} \right)^2 - \frac{\tau_C^2}{4\alpha} \geq \varphi_C \right\},$$

thus, results are formulated in the next set

$$K_C(h_1) = \left\{ C(t) \geq C_{\text{inf}} = \sqrt{\frac{\varphi_C}{\alpha} + \frac{\tau_C^2}{4\alpha^2}} - \frac{\tau_C}{2\alpha} \right\},$$

it is evident that if the CAR-T cell therapy dose is zero, then the lower bound $C_{\text{inf}} = 0$. Therefore, as long as therapy is being administered to the patient, a concentration of CAR-T cells can be expected in the system. Now, the following localizing function is applied in order to find the upper bound:

$$h_2 = C + L,$$

the Lie derivative is computed as follows:

$$L_f h_2 = \varphi_C - (\alpha - \rho_C)LC - (\alpha - \rho_C)C^2 - \tau_C C - \rho_L \tau_L L^2 + \rho_L L,$$

and the next constraint is established,

$$\alpha > \rho_C,$$

note that the latter always holds, parameter $\rho_C$ is a proportion of $\alpha$ (see condition (3) and Table 1). Thus, set $S(h_2) = \left\{ L_f h_2 = 0 \right\}$ can be written as indicated below,

$$S(h_2) = \left\{ \tau_C C = \varphi_C - (\alpha - \rho_C)LC - (\alpha - \rho_C)C^2 - \rho_L \tau_L L^2 + \rho_L L \right\},$$

now, by substituting $C = h_2 - L$ into the set, completing the square and performing basic arithmetic operations, we get the following,

$$S(h_2) = \left\{ h_2 = \frac{\varphi_C}{\tau_C} + \frac{(\rho_L + \tau_C)^2}{4\rho_L \tau_C \tau_L} - \frac{1}{\tau_C} f(L, C) \right\},$$

where

$$f(L, C) = (\alpha - \rho_C)LC + (\alpha - \rho_C)C^2 + \rho_L \tau_L \left( L - \frac{\rho_L + \tau_C}{2\rho_L \tau_L} \right)^2,$$

and the localizing domain of $h_2 = C + L$ is given as follows:

$$K(h_2) = \left\{ C(t) + L(t) \leq \frac{\varphi_C}{\tau_C} + \frac{(\rho_L + \tau_C)^2}{4\rho_L \tau_C \tau_L} \right\},$$

therefore, one can estimate the next upper bound of the CAR-T cells

$$K_C(h_2) = \left\{ C(t) \leq C_{\text{sup}} = \frac{\varphi_C}{\tau_C} + \frac{(\rho_L + \tau_C)^2}{4\rho_L \tau_C \tau_L} \right\}.$$

Now, by applying the next localizing function, one can determine an ultimate upper bound for the ALL cancer cells population:

$$h_3 = L,$$

whose Lie derivative is given by

$$L_f h_3 = \rho_L L(1 - \tau_L L) - \alpha LC,$$

thus, set $S(h_3) = \left\{ L_f h_3 = 0 \right\}$ is given as follows:

$$S(h_3) = \left\{ L = \frac{1}{\tau_L} - \frac{\alpha}{\rho_L \tau_L} C \right\} \cup \{L = 0\},$$

at this step, one needs to apply the Iterative Theorem as shown below,

$$S(h_3) \cap K_C(h_1) \subset \left\{ L \leq \frac{1}{\tau_L} - \frac{\alpha}{\rho_L \tau_L} C_{\text{inf}} \right\},$$

from the latter, the next upper bound is determined

$$K_L(h_3) = \left\{ L(t) \leq L_{\text{sup}} = \frac{1}{\tau_L} - \frac{\alpha}{\rho_L \tau_L} C_{\text{inf}} \right\}.$$

Results shown above allow us to formulate the next statement:

**Theorem 4.** *Localizing Domain. If condition (3) holds, then lower and upper bounds for CAR-T cells $[C(t)]$ and ALL cancer cells $[L(t)]$ populations are given in the following localizing domain:*

$$\Gamma_{CL} = \Gamma_C \cap \Gamma_L,$$

*where*

$$CAR\text{-}T\ cells \quad : \quad \Gamma_C = \left\{ C_{\mathrm{inf}} \leq C(t) \leq C_{\mathrm{sup}} \right\},$$
$$ALL\ cancer\ cells \quad : \quad \Gamma_L = \left\{ 0 \leq L(t) \leq L_{\mathrm{sup}} \right\}.$$

Nonexistence conditions may be computed as a secondary result of Theorem 4 by formulating the following constraint:

$$L_{\mathrm{sup}} < 0,$$

which is rewritten as follows:

$$\frac{1}{\tau_L} - \frac{\alpha}{\rho_L \tau_L} \left( \sqrt{\frac{\varphi_C}{\alpha} + \frac{\tau_C^2}{4\alpha^2}} - \frac{\tau_C}{2\alpha} \right) < 0, \tag{5}$$

and solved for the CAR-T cells therapy parameter $[\varphi_C]$

$$\varphi_C > \varphi_{CART} = \frac{\rho_L}{\alpha}(\rho_L + \tau_C), \tag{6}$$

thus we are able to establish the next statement.

**Corollary 1.** *Nonexistence. If condition (6) holds, then there are no compact invariant sets for the ALL and CAR-T cells therapy system outside the plane $L = 0$.*

*4.2. Eradication Conditions*

Note that the fulfilment of the nonexistence condition (6) does not imply the leukaemia cancer cells eradication described by the system (1)–(2). Therefore, we apply Lyapunov's Direct Method in order to derive sufficient conditions that can allow us to establish global asymptotic stability of the tumour-free equilibrium point and ensure the complete ALL cancer cells eradication. Thus, let us analyse the following Lyapunov candidate function:

$$V = L,$$

and compute its temporal derivative as follows:

$$\dot{V} = \rho_L L - \rho_L \tau_L L^2 - \alpha LC,$$

thus, one can formulate an upper bound by evaluating the function at the localizing domain, i.e., $\dot{V}\big|_{\Gamma_{CL}}$, and get the next result

$$\dot{V} \leq (\rho_L - \alpha C_{\mathrm{inf}}) L \leq 0,$$

thus, if the next condition holds

$$\rho_L - \alpha \sqrt{\frac{\varphi_C}{\alpha} + \frac{\tau_C^2}{4\alpha^2}} + \frac{\tau_C}{2} < 0, \tag{7}$$

then $\dot{V}$ is negative definite, i.e., $\dot{V}(0) = 0$ and $\dot{V} < 0\ \forall\ L > 0$. Thus, by solving (7) for the CAR-T cells therapy parameter $[\varphi_C]$ we get the following,

$$\varphi_C > \varphi_{CART}.$$

Therefore, in this particular case, nonexistence and global asymptotic stability conditions are the same for this mathematical model describing cancer evolution. Results shown in this section allow us to formulate the next statement.

**Theorem 5.** *ALL Cancer Cells Eradication. If the CAR-T cells therapy dose $[\varphi_C]$ fulfils condition (6), then the ALL cancer cells population $[L(t)]$ described by the system (1) and (2) is completely eradicated for any initial tumour size $[L(0)]$. Hence,*

$$\lim_{t \to \infty} L(t) = 0 \; \forall \; L(0) > 0 \; \Leftrightarrow \; \varphi_C > \varphi_{CART},$$

*and the tumour-free equilibrium point (4) $[(C_0^*, L_0^*)]$ is globally asymptotically stable.*

If condition from Theorem 5 holds, then $L(t) = 0$ and Equation (1) is rewritten as follows:

$$\dot{C} = -(\alpha - \rho_C)C^2 - \tau_C C + \varphi_C,$$

which has a unique biologically meaningful (non-negative) equilibrium given by $C_0^*$ that is globally asymptotically stable, i.e., $\lim_{t \to \infty} C(t) = C_0^*$. Additionally, once the ALL cancer cells population is eradicated then the CAR-T cells therapy can be stopped $[\varphi_C = 0]$ and $C_0^* = 0$. This implies that the CAR-T cells population will eventually be depleted in the patient, i.e., $\lim_{t \to \infty} C(t) = 0$. Results determined in this section are illustrated and discussed below by means of in silico experimentations.

## 5. Discussion and In Silico Experimentation

In this section, our mathematical results are explored by means of the so-called in silico experimentation [30,31] in the form of several numerical simulations under different assumptions regarding parameter values, initial tumour burdens, and therapeutic doses.

The ALL cancer cells eradication condition (6) formulated on the immunotherapy treatment $[\varphi_C]$ (see Corollary 1 and Theorem 5) may be fulfilled for diverse scenarios as it is written in terms of the following three parameters: $\rho_L$, the leukaemia cancer cells growth rate; $\alpha$, the killing efficacy rate of CAR-T cells; and $\tau_C$, its natural death rate. From previous works [30,31,37–40], we have found that both the cancer cells growth rate and the killing efficacy of the treatment are consistently involved in tumour clearance conditions. Therefore, scientifically formulating a personalised treatment strategy for each patient requires accurately fitting the values of these two parameters. However, this may be difficult to achieve: clinical trials do not always provide all the required information to estimate these parameters as it is not the main purpose of these studies.

One of particular interest is that of Lee et al. [11], where they performed a phase 1 dose-escalation trial on CAR-T cells for the treatment of B-ALL on 21 patients concluding that this therapy is feasible, safe and mediates potent anti-leukaemic activity in children and young adults. Further, the authors provided important information concerning dose, intervals of the therapy application, thresholds for cancer remission and CAR-T cells detectability in the peripheral blood, overall toxicity and a period for CR. Our main interest relies on the CR period and thresholds for remission and clinical detectability. These are going to be applied with the complete eradication threshold (see Assumption 1) to estimate a set of values for the $\alpha$ parameter, as well as to discuss the feasibility of the system (1)–(2) to reproduce the ultimate ALL cancer dynamics under CAR-T cell therapy.

First, let us summarised Lee et al. methods and results [11]. Using a standard protocol guideline of $3 + 3$ to establish the maximum tolerated dose [41], therapy was infused on days 0 and 7. Expansion of CAR-T cells occurs during the first 2 weeks, followed by a rapid CAR-T cell contraction. Dose 1 was $1 \times 10^6$ per kg and dose 2 was $3 \times 10^6$ per kg, both of CAR-transduced T cells. Protocol-prescribed doses were successfully produced for 19 of 21 patients for a 90% feasibility rate. Of the 14 responding patients, 12 had undetectable circulating B cells after treatment between days 14 and 28 or shortly thereafter. Information concerning the weight of each patient was not registered in their results.

Therefore, numerical simulations should illustrate that the B-ALL cancer cells population is at least below the remission threshold by day 28 after the initial application of the described two-dose therapy protocol. Nonetheless, we aim for all solutions to be below the eradication value of 1 cancer cell in our in silico experimentation process. First, let us

consider the following concerning weight, initial conditions and other parameters shown in Table 1 to estimate the necessary values of $\alpha$ to achieve CR:

1. Patient weight is increased by 10 kg in each iteration going from 10 to 100 kg.
2. Six initial tumour burdens of $10^5 - 10^{10}$ are set simultaneously for each weight.
3. The maximum tumour carrying capacity $\left[\tau_L^{-1}\right]$ is set to $4.64 \times 10^{11}$ cells. As shown in Table 1, this is the lower value for this parameter as it is taking into account that B-ALL occurs mostly in children [3] with a median age of diagnosis of 17 years old [4].
4. For the remaining parameters, different combinations are explored by setting one value for the mitosis stimulation rate of CAR-T cells due to encounters with B-ALL cancer cells $[\rho_C = 0.2\alpha]$; two values for the natural death rate of CAR-T cells $[\tau_C = 1/14, 1/30]$; and three values for the cancer cells growth rate $[\rho_L = 1/60, 1/40, 1/20]$. The latter is determined from the main results shown in [23,24].

Numerical simulations were performed by applying Euler's method $[x_{i+1} = x_i + f(x)\Delta_t$, see [42] at Section 1.7] to solve system (1) and (2) with a step size $\Delta_t = 1 \times 10^{-5}$ to further reduce the intrinsic error in Euler's solutions. Table 2 summarises our results for the set parameter values.

**Table 2.** Estimated values for the $\alpha$ parameter in order to achieve CR in a patient by day 28 after the two doses of CAR-T cells therapy on days 0 [dose 1] and 7 [dose 2]. As the total dose applied depends on the weight of the patient, values are estimated for 10 kg to 100 kg. In silico experimentations were performed for two values of the natural death rate of CAR-T cells $[\tau_C]$, and three values of the cancer cells growth rate $[\rho_L]$. For the maximum tumour carrying capacity and the mitosis stimulation rate of CAR-T cells only one value was considered. Note that all results of $\alpha$ are in the order of $10^{-7}$.

| Weight | Dose 1 | Dose 2 | Total | Estimated Value of $\alpha$ $\left[\times 10^{-7}\right]$ | | | | | |
| | $1 \times 10^6$ | $3 \times 10^6$ | | $\rho_L = 1/60$ | | $\rho_L = 1/40$ | | $\rho_L = 1/20$ | |
| kg | cells $\times$ kg | cells $\times$ kg | cells | $\tau_C = 1/30$ | $\tau_C = 1/14$ | $\tau_C = 1/30$ | $\tau_C = 1/14$ | $\tau_C = 1/30$ | $\tau_C = 1/14$ |
|---|---|---|---|---|---|---|---|---|---|
| 10 | $1 \times 10^7$ | $3 \times 10^7$ | $4 \times 10^7$ | 7.147 | 7.858 | 7.430 | 8.158 | 8.312 | 9.001 |
| 20 | $2 \times 10^7$ | $6 \times 10^7$ | $8 \times 10^7$ | 4.201 | 4.594 | 4.357 | 4.759 | 4.843 | 5.221 |
| 30 | $3 \times 10^7$ | $9 \times 10^7$ | $1.1 \times 10^8$ | 3.064 | 3.340 | 3.174 | 3.456 | 3.515 | 3.781 |
| 40 | $4 \times 10^7$ | $1.2 \times 10^8$ | $1.6 \times 10^8$ | 2.441 | 2.657 | 2.527 | 2.747 | 2.793 | 2.999 |
| 50 | $5 \times 10^7$ | $1.5 \times 10^8$ | $2 \times 10^8$ | 2.043 | 2.220 | 2.114 | 2.294 | 2.332 | 2.502 |
| 60 | $6 \times 10^7$ | $1.8 \times 10^8$ | $2.4 \times 10^8$ | 1.763 | 1.914 | 1.824 | 1.978 | 2.010 | 2.154 |
| 70 | $7 \times 10^7$ | $2.1 \times 10^8$ | $2.8 \times 10^8$ | 1.555 | 1.687 | 1.608 | 1.742 | 1.770 | 1.896 |
| 80 | $8 \times 10^7$ | $2.4 \times 10^8$ | $3.2 \times 10^8$ | 1.393 | 1.511 | 1.440 | 1.560 | 1.585 | 1.696 |
| 90 | $9 \times 10^7$ | $2.7 \times 10^8$ | $3.6 \times 10^8$ | 1.264 | 1.369 | 1.306 | 1.413 | 1.436 | 1.536 |
| 100 | $1 \times 10^8$ | $3 \times 10^8$ | $4 \times 10^8$ | 1.157 | 1.253 | 1.196 | 1.293 | 1.314 | 1.405 |

The killing efficacy of T cells has been the subject of interest in several works regarding the modelling of cancer evolution and immune response, and the significant results are shown in Table 3.

**Table 3.** Estimated values of effector T cells efficacy on recognising and eliminating cancer cells in other researches concerning the mathematical modelling of both solid and non-solid tumours by means of first-order ODEs.

| Value | Source |
|---|---|
| $9.999 \times 10^{-7}$ | [43,44] |
| $9.530 \times 10^{-7}$ | [45] |
| $1.269 \times 10^{-7}$ | [46,47] |
| $1 \times 10^{-9}$ | [48] |
| $3.422 \times 10^{-10}$ | [49] |
| $3.410 \times 10^{-10}$ | [50] |
| $5.84 \times 10^{-11}$ | [23,24] |
| $5 \times 10^{-11}$ | [51] |

It is evident that results from Table 2 are close to those presented by Kirschner et al. [43,44] and de Pillis et al. [46,47] when applying an adoptive immunotherapy treatment in the form of tumour infiltrating lymphocytes and Kronik et al. [45] in their clinical trial testing an allogeneic prostate cancer whole-cell vaccine. Therefore, our estimated values of $\alpha$ are consistent with those find in the literature.

Now, to further continue with our in silico experimentation in the ALL and CAR-T cells system (1) and (2), a hypothetical 60 kg patient is selected as this is the average weight in both female and male children of 17 years old [52]. The particular characteristics for this patient are set as follows: a CAR-T cells lifetime $\left[\tau_C^{-1}\right]$ of 14 days [23,53], and a rapidly growing tumour, i.e., $\rho_L = 1/20$. Therefore, the killing efficacy of the therapy $[\alpha]$ should be $2.154 \times 10^{-7}$ to achieve CR by day 28, see Table 2. Further, three initial $[L(0)]$ B-ALL tumour burdens given by $10^7$, $10^8$ and $10^9$ cells are investigated as most of the patients in the clinical trial of Lee et al. are between these values, see Figure 1 at [11].

First, the four CAR-T cells therapy administration protocols are illustrated in Figure 1. The one defined in the clinical trial performed by Lee et al. [11] is compared with three protocols of our own. Then, Figures 2–5 illustrate that CR is achieved by day 28 and CAR-T cells are below the threshold of clinical detectability by day 77. Further, general information such as initial conditions [point $\circ$ at $t = 0$], remission and detectability thresholds $[- -$ line at $10^4$ cells], and the eradication threshold $[- \times$ line at 1 cell] are indicated in every corresponding panel.

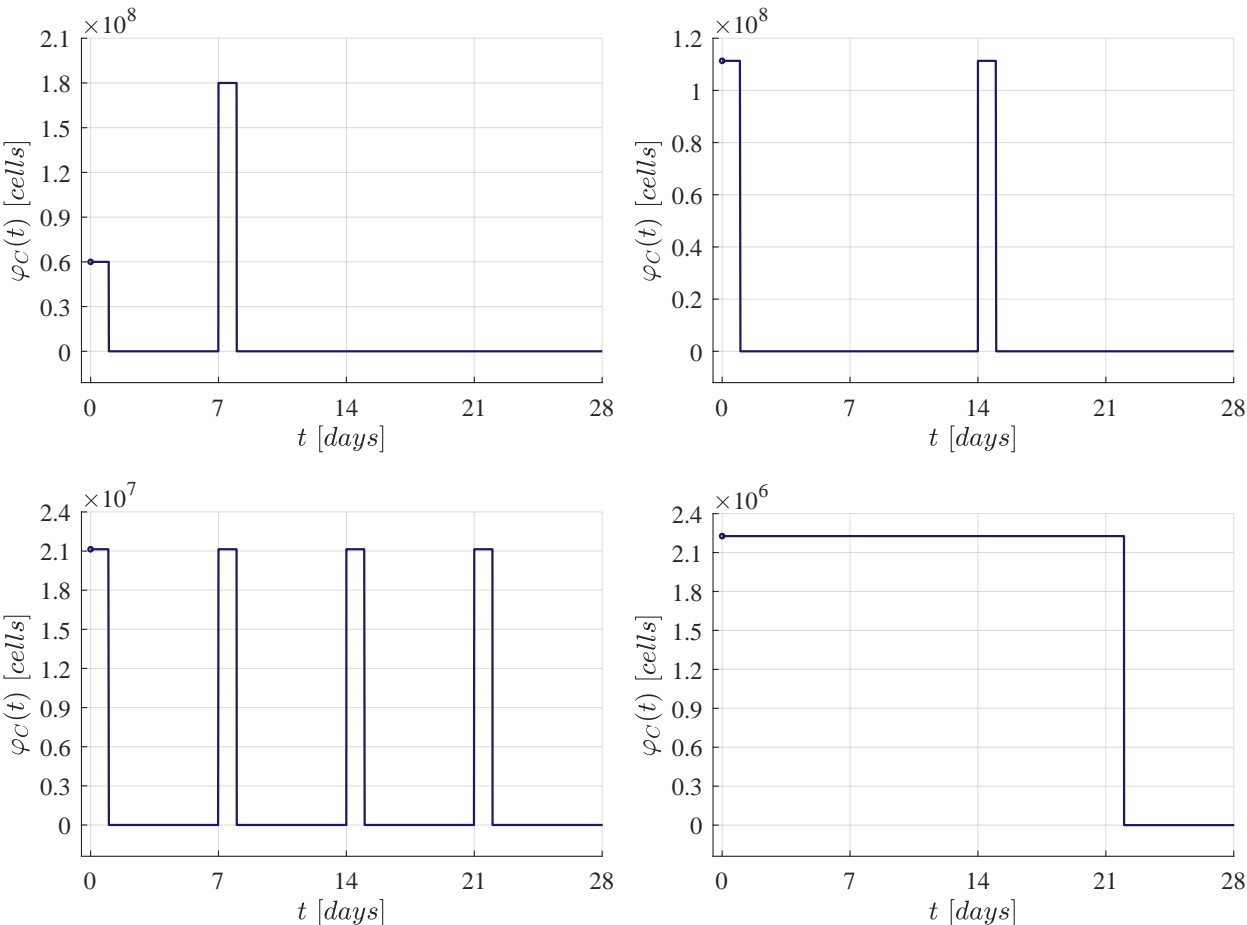

**Figure 1.** CAR-T cells therapy administration protocols for CR by day 28. Upper left panel: Dose 1 of $6 \times 10^7$ at day 0, and dose 2 of $1.8 \times 10^8$ at day 7, this protocol was applied in the Lee et al. [11] clinical trial. The following results were determined by means of in silico experimentation and the ALL cancer cells eradication condition (6). Upper right panel: Two doses of approximately $1.114 \times 10^8$ at days 0 and 14. Lower left panel: Four doses of approximately $2.115 \times 10^7$ at days 0, 7, 14 and 21. Lower right panel: Daily doses of approximately $2.227 \times 10^6$ from days 0 to 21.

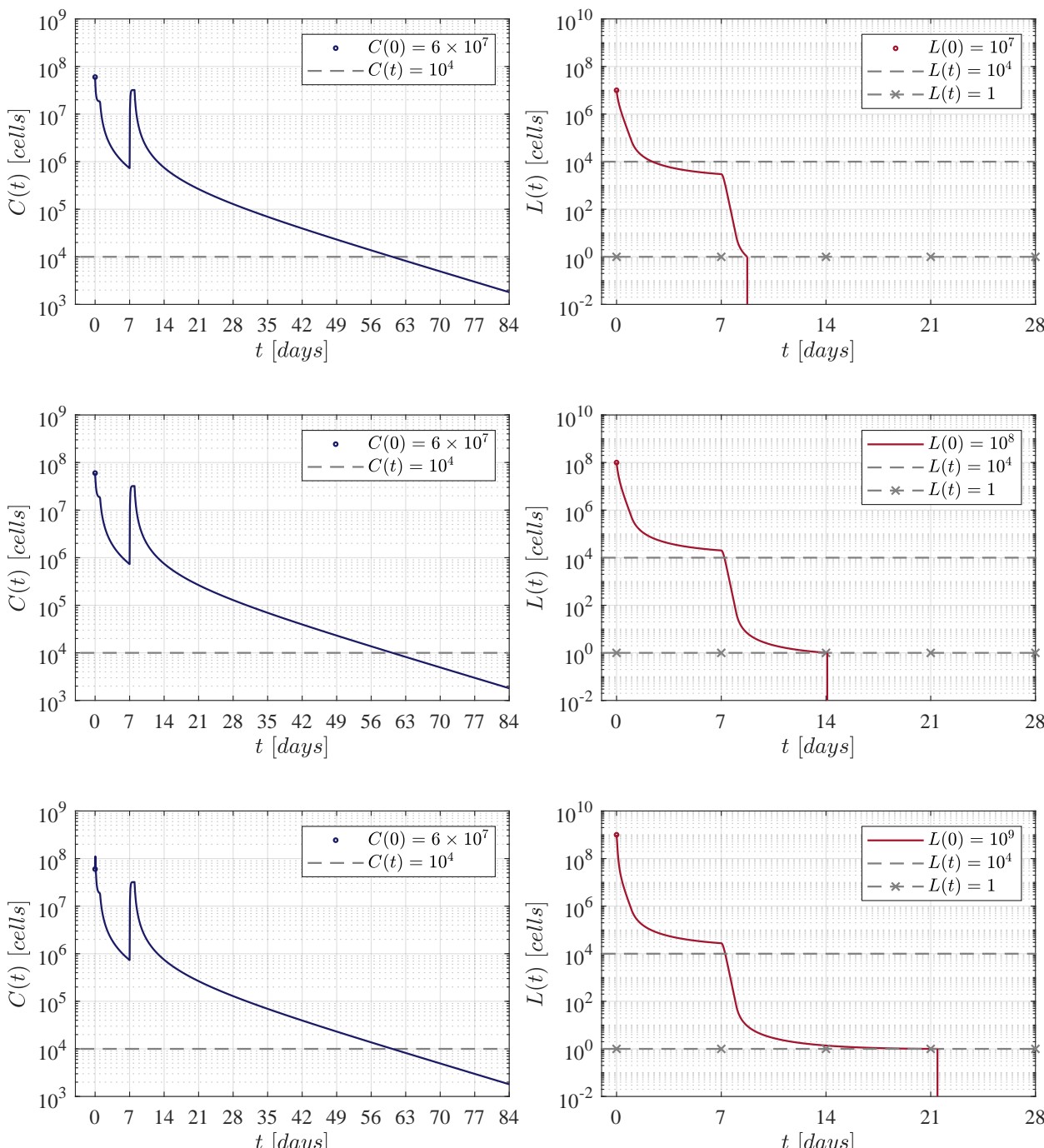

**Figure 2.** Results determined when applying the Lee et al. [11] therapy administration protocol, i.e., an initial dose of $6 \times 10^7$ CAR-T cells at day 0, and a second one of $1.8 \times 10^8$ at day 7.

Figure 2 illustrates results when applying the therapy administration protocol designed by Lee et al. [11]. The protocol is given as follows: an initial dose of $6 \times 10^7$ at day 0 and a second dose of $1.8 \times 10^8$ at day 7, which implies that a total of $2.4 \times 10^8$ CAR-T cells are infused by the end of the treatment. One can see that as the initial leukaemia cancer cells burden increases, then the population takes a few more days to go below the threshold for tumour eradication. The time-evolution of the CAR-T cells remains almost the same through the three iterations with a small peak (lower left panel) due to the high initial tumour burden that stimulates clonal expansion. The latter is observed again in

Figures 3–5. Further, by day 63 the population is already below the detectability threshold of $10^4$ cells in this particular case.

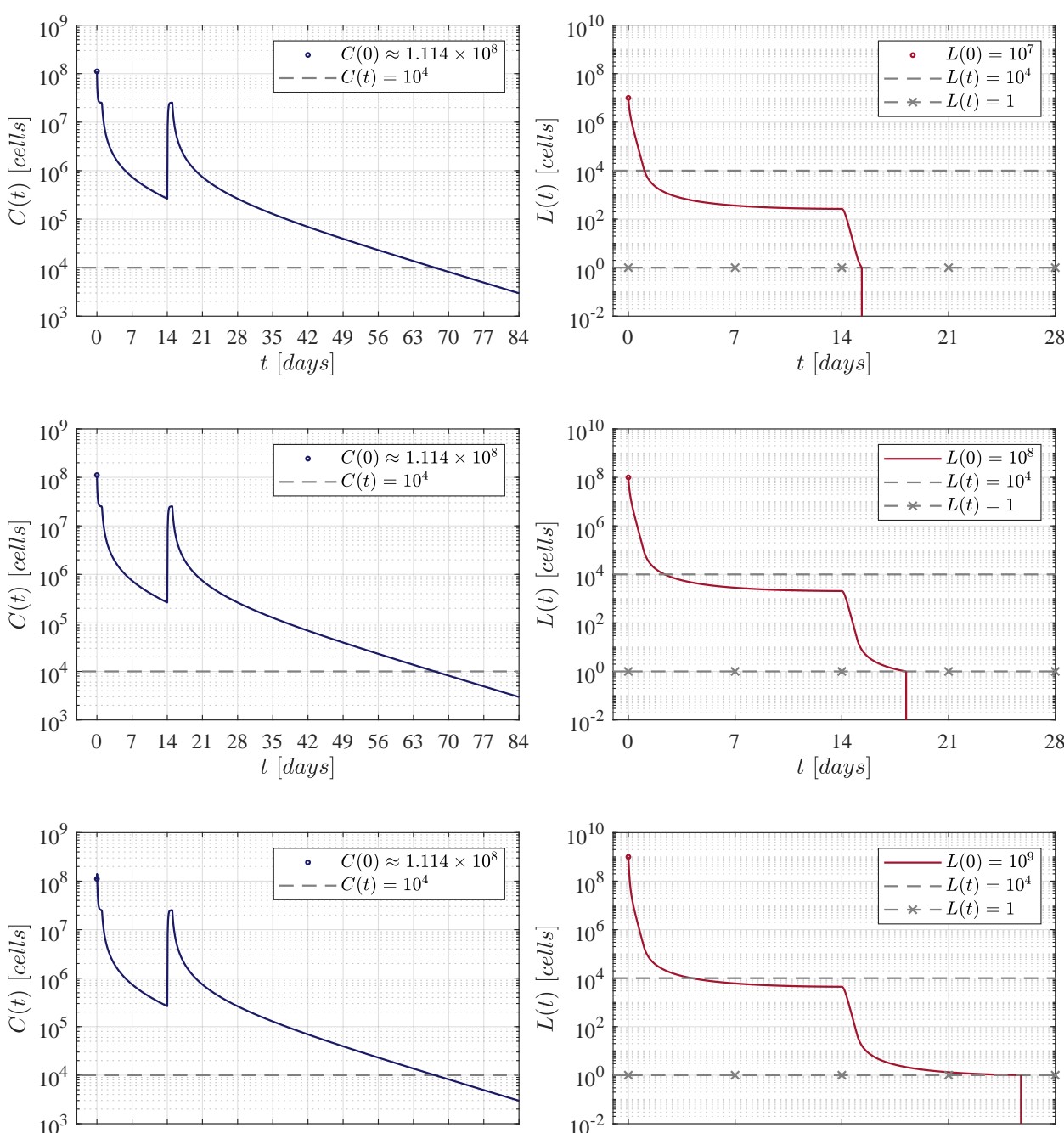

**Figure 3.** The fortnight administration protocol. Two applications were applied, the first one at day 0 and the second at day 14. In silico experimentations allow us to determine that in this approach a dose of 111, 337, 711 CAR-T cells was needed to achieve CR by day 28.

Figures 3–5 illustrate results of the three CAR-T cells therapy administration protocols formulated in the form of impulse trains with an amplitude (dose) given by a proportion of the ALL cancer cells eradication condition (6), whose value is computed as follows:

$$\varphi_C > \varphi_{CART} = \frac{\rho_L}{\alpha}(\rho_L + \tau_C) = 28,187 \text{ CAR-T cells}, \tag{8}$$

where $\rho_L = 1/20$, $\alpha = 2.154 \times 10^{-7}$, and $\tau_C = 1/14$.

In the first iteration shown in Figure 3, two applications of the therapy are applied at days 0 and 14, i.e., two weeks apart. The necessary dose for each application was estimated as 111, 337, 711 CAR-T cells for the three initial tumour burdens. The latter implies that a total of 222, 675, 422 cells were infused. This value is very close to that obtained by Lee et al. in their clinical trial. In this case, CAR-T cells are below the detectability threshold by day 70.

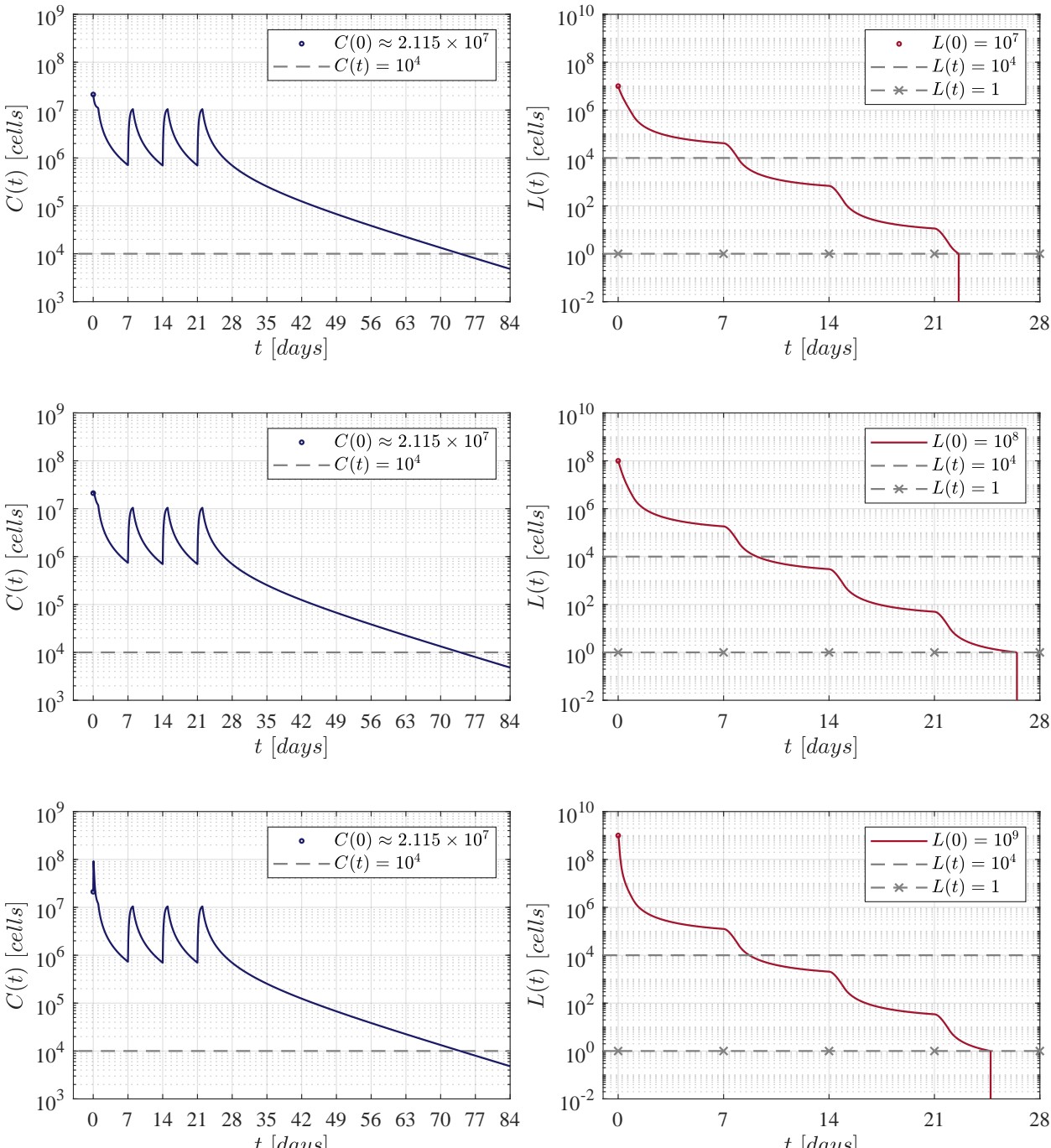

**Figure 4.** The weekly administration protocol. A total of four applications were applied at days 0, 7, 14 and 21. In silico experimentations allow us to determine that a dose of 21, 140, 072 CAR-T cells was necessary to achieve CR by day 28.

The second iteration considered weekly applications at days 0, 7, 14 and 21 as it is illustrated in Figure 4. In this case, numerical simulations have shown that a dose of 21, 140, 072 CAR-T cells was needed for each application. Therefore, the total amount of cells infused was significantly lower with a final value of 84, 560, 288 CAR-T cells, and the population is below the detectability threshold by day 77.

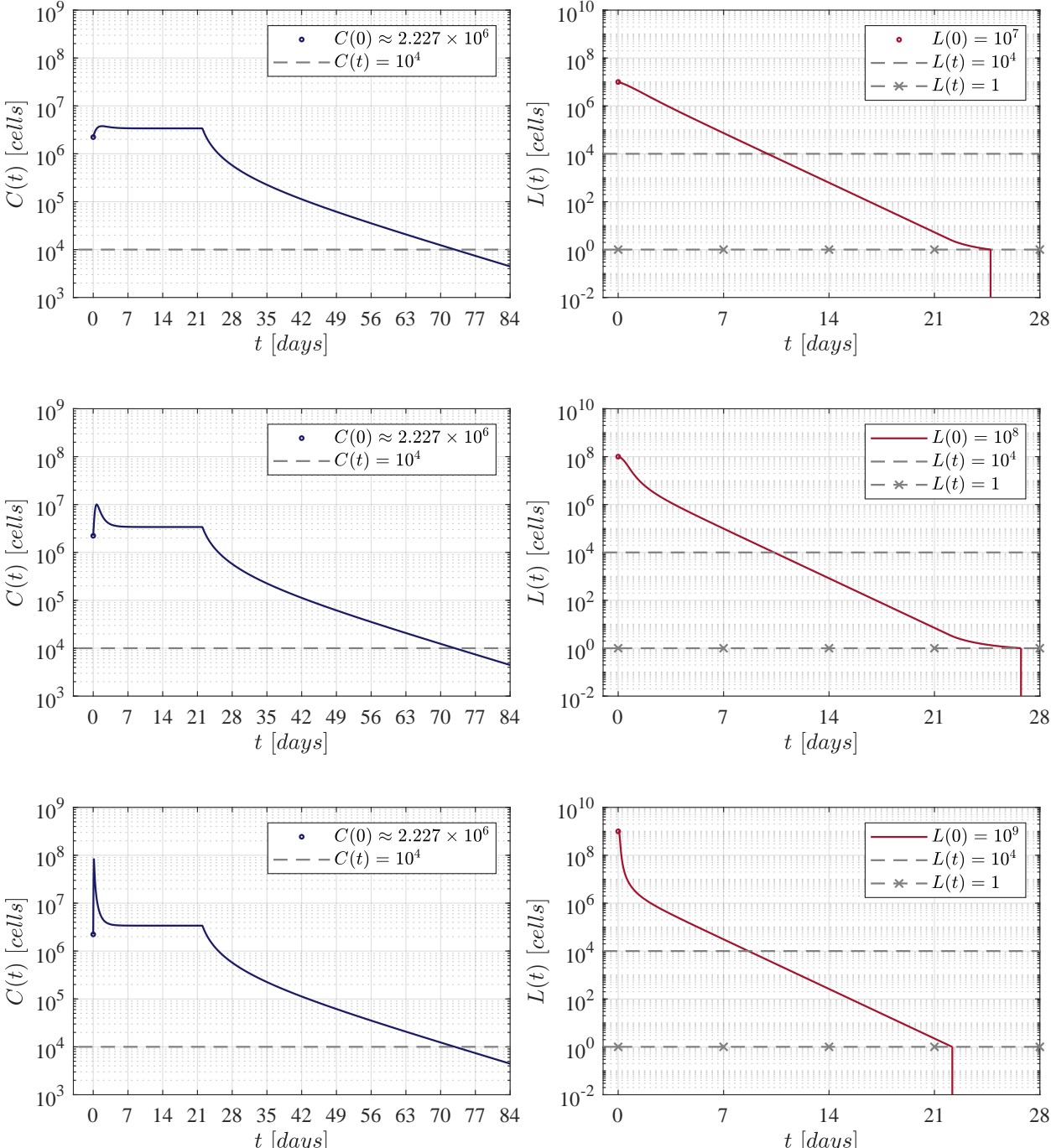

**Figure 5.** The daily administration protocol. In this case we aim for CR by day 28 by means of daily therapy applications from day 0 to day 21 as it was the last day of application in the weekly protocol illustrated in Figure 4. The latter allows for one week of rest between the last dose and the peripheral blood test indicated in Lee et al. [11] at day 28.

For the third and final iteration, the therapy was applied every day, and as it is seen in Figure 5, the daily dose was determined to be 2, 226, 755 CAR-T cells, this means that the final amount by the end of day 21 was a total of 48, 988, 610 cells. In this iteration, CAR-T cells are below the detectability threshold by day 77. However, to the best of our knowledge, we should note that this may not be a suitable administration protocol for the treatment in a real-life scenario. Nonetheless, it is evident that shortening the period of the applications ultimately decreases both the dose and the final amount of therapy that needs to be infused into the patient in order to achieve CR.

Concerning the dynamics of CAR-T cells, one can see in Figures 2–5 that solutions of $C(t)$ go below the threshold of clinical detectability of $10^4$ cells between days 56 and 77 even though the B-ALL cancer cells population is eliminated by day 28. The latter could be explained by the self-stimulation term given by $\rho_C C \times C$ in Equation (1), indicating that the given value should be lower than the one considered in this and previous works. Nonetheless, further information is required in order to accurately fit parameter $\rho_C$.

Although the results illustrated and discussed in this section are only for the established scenario, note that all sixty values computed for $\alpha$ in Table 2 were estimated in this form. The latter implies that CR by day 28 is achieved in all other cases, and it is straight forward to illustrate that the B-ALL cancer cells population $[L(t)]$ goes below the threshold of tumour eradication for any initial burden in the range of $10^5$–$10^{10}$ cells, for any weight going from 10 to 100 kg, and for every combination on the parameter values.

## 6. Conclusions

In silico experimentation provides a powerful tool that could potentially be applied in the design of personalised strategies for cancer therapy administration protocols. Numerical simulations may help us to observe beyond the thresholds of clinical detectability in current cancer imaging technologies. Furthermore, this approach allows researchers to explore multiple scenarios at the same time when taking into account different possibilities on the health of the patient. The latter due to the solutions illustrating the efficacy of the cancer therapy in both short- and long-term, information that can be used to decide the best treatment strategy for each case.

In this work, we were able to formulate three CAR-T cell therapy administration protocols (see Figure 1) and compare them with the overall results reported by Lee et al. when exploring a phase 1 dose escalation trial for the treatment of B-ALL. As illustrated in Figures 2–5, CR was achieved in all strategies where different doses and intervals of application were tested. The amount of each dose was computed as a proportion of the ALL cancer cells eradication condition (6), this was established in Corollary 1 and Theorem 5 by means of the LCIS and Lyapunov's direct methods. Therefore, we can conclude that the combined application of nonlinear systems theory and systems biology with in silico experimentation can provide useful information in the designing of cancer therapy administration protocols.

Regarding the dose of the therapy, Figures 3–5 gradually illustrate that more applications of the therapy decrease the total amount of CAR-T cells that needs to be infused into the patient for CR. Nonetheless, the feasibility of these strategies remains as an open question that needs to be further investigated and validated by clinical trials.

Finally, our results indicate that factors such as tumour growth, immune response and the efficacy of the therapy in eliminating malignant cells, among many others, can be translated into specific parameter values that ultimately yield different tumour dynamics. Therefore, by accurately estimating these data, mathematical models that can better describe real-life scenarios of cancer evolution of either solid or non-solid tumours would eventually be formulated and validated at the same time. In the specific case of leukaemia, blood cell counts can be determined by means of peripheral blood flow cytometry which is performed by Lee et al. in their clinical trial. Nonetheless, measurements are acquired for a very limited number of days, and they are not identified for each patient. However, by incorporating daily blood cell counts to these types of clinical trials to properly mea-

sured each population, these data can be modelled through genetic algorithms focusing on objective functions such as those from growth laws. Furthermore, parameter values can be estimated with 95% confidence interval through nonlinear curve-fitting [54].

**Author Contributions:** Conceptualisation, P.A.V. and Y.S.; methodology, P.A.V. and L.N.C.; software, P.A.V. and C.P.; validation, Y.S. and L.N.C.; formal analysis, P.A.V.; investigation, Y.S. and C.P.; resources, P.A.V. and L.N.C.; writing—original draft preparation, P.A.V. and Y.S.; writing—review and editing, L.N.C. and C.P.; visualisation, P.A.V. and L.N.C.; project administration, P.A.V. All authors have read and agreed to the published version of the manuscript.

**Funding:** This research received no external funding.

**Institutional Review Board Statement:** Not applicable.

**Informed Consent Statement:** Not applicable.

**Data Availability Statement:** Data is contained within the article.

**Acknowledgments:** This research is fulfilled within the TecNM project number 9951.21-P 'Computational modelling and in silico experimentation applied to the analysis and control of biological systems' (Modelizado computacional y experimentos in silico aplicados al análisis y control de sistemas biológicos).

**Conflicts of Interest:** The authors declare no conflict of interest.

## Abbreviations

The following abbreviations are used in this manuscript:

| | |
|---|---|
| ALL | Acute Lymphocytic Leukaemia |
| CAR | Chimeric Antigen Receptor |
| CR | Complete Response |
| LCIS | Localization of Compact Invariant Sets |
| ODEs | Ordinary Differential Equations |

## Appendix A. Equilibria, Persistence, Existence and Uniqueness

In this appendix, we further explore the overall dynamics of the ALL and CAR-T cells system (1) and (2). First, let us determine the equilibrium points of Equations (1) and (2) by equating them to zero as follows:

$$\begin{aligned} f_1(t, C, L) &= \rho_C(L + C)C - \tau_C C - \alpha C^2 + \varphi_C = 0, \\ f_2(t, C, L) &= (\rho_L - \rho_L \tau_L L - \alpha C)L = 0, \end{aligned}$$

and by solving them for each state variable $[C(t), L(t)]$ one can determine the next four equilibrium points

$$\begin{aligned} &(C_0^*, L_0^*), \\ &(C_1^*, L_1^*), \\ &(C_2^*, L_2^*), \\ &(C_3^*, L_0^*), \end{aligned}$$

$$\text{(A1)}$$
$$\text{(A2)}$$
$$\text{(A3)}$$

where

$$C_0^* = \frac{\sqrt{\tau_C^2 + 4\varphi_C(\alpha - \rho_C)} - \tau_C}{2(\alpha - \rho_C)},$$

$$C_1^* = \frac{\rho_L(\rho_C - \tau_C\tau_L) + \sqrt{\rho_L^2(\tau_C\tau_L - \rho_C)^2 + 4\varphi_C\rho_L\tau_L[\alpha\rho_C + \tau_L\rho_L(\alpha - \rho_C)]}}{2[\alpha\rho_C + \tau_L\rho_L(\alpha - \rho_C)]},$$

$$C_2^* = \frac{\rho_L(\rho_C - \tau_C\tau_L) - \sqrt{\rho_L^2(\tau_C\tau_L - \rho_C)^2 + 4\varphi_C\rho_L\tau_L[\alpha\rho_C + \tau_L\rho_L(\alpha - \rho_C)]}}{2[\alpha\rho_C + \tau_L\rho_L(\alpha - \rho_C)]},$$

$$C_3^* = -\frac{\tau_C + \sqrt{\tau_C^2 + 4\varphi_C(\alpha - \rho_C)}}{2(\alpha - \rho_C)},$$

$$L_0^* = 0,$$

$$L_1^* = \frac{\alpha(\tau_C\tau_L + \rho_C) + 2\tau_L\rho_L(\alpha - \rho_C) - \alpha\sqrt{(\tau_C\tau_L - \rho_C)^2 + 4\varphi_C\tau_L\left[\alpha\rho_C\rho_L^{-1} + \tau_L(\alpha - \rho_C)\right]}}{2\tau_L[\alpha\rho_C + \tau_L\rho_L(\alpha - \rho_C)]},$$

$$L_2^* = \frac{\alpha(\tau_C\tau_L + \rho_C) + 2\tau_L\rho_L(\alpha - \rho_C) + \alpha\sqrt{(\tau_C\tau_L - \rho_C)^2 + 4\varphi_C\tau_L\left[\alpha\rho_C\rho_L^{-1} + \tau_L(\alpha - \rho_C)\right]}}{2\tau_L[\alpha\rho_C + \tau_L\rho_L(\alpha - \rho_C)]},$$

remembering that $(C_0^*, L_0^*)$ was introduced at the end of Section 3 with the tumour-free equilibrium point (4), and $\alpha > \rho_C$ as condition (3) always holds. Therefore, the following assertions can be made.

1. There are two tumour-free equilibrium points, from which (4) is non-negative $[C_0^* \geq 0, L_0^* = 0]$ and (A3) is negative $[C_3^* < 0, L_0^* = 0]$.
2. There are three biologically feasible equilibria given by (4), (A1) and (A2).

Now, given that $f_i(t, x)$, $i = 1, 2$; and $x = [C, L]^T$, then the Jacobian matrix (see [42] at Section 7.4) $[\partial f / \partial x](t, x)$ of (1) and (2) is computed as indicated below

$$J = \begin{bmatrix} \rho_C L + 2\rho_C C - \tau_C - 2\alpha C & \rho_C C \\ -\alpha L & \rho_L - 2\tau_L\rho_L L - \alpha C \end{bmatrix}, \tag{A4}$$

and it is evident that both $f_i(t, x)$ and $[\partial f / \partial x](t, x)$ are continuous and exist on the domain $\Omega = [t_0, t_1] \times \Gamma_{CL}$ with $[t_0, t_1] \in [t_0, \infty)$ and $\Gamma_{CL} \subset \mathbf{R}_{+,0}^2$.

Regarding existence and uniqueness, the latter implies that $f_i(t, x)$ is locally Lipschitz in $x$ on $\Omega$, see Lemma 3.2 by Khalil in [27] at Section 3.1. Further, it is possible to ensure that each element of (A4) is bounded by Theorem 4, and we conclude the following.

**Theorem A1.** *Existence and uniqueness. There is a Lipschitz constant $\ell \geq 0$ such that $\|[\partial f / \partial x](t, x)\| \leq \ell$ on $\Omega$. Then, $f_i(t, x)$ satisfies the Lipschitz condition*

$$\|f(t, x_1) - f(t, x_2)\| \leq \ell\|x_1 - x_2\|,$$

*and there exists some $\delta > 0$ such that the ALL and CAR-T cells system (1) and (2), given as $\dot{x} = f_i(t, x)$ with $x(t_0) = x_0$, has a unique solution over $[t_0, t_0 + \delta]$.*

In order to derive necessary conditions for ALL cancer cells persistence in the solutions of system (1) and (2), the local stability of the tumour-free equilibrium point (4) should be investigated. Hence, let us evaluate this point at the Jacobian matrix (A4) as follows:

$$J|_{(C_0^*, L_0^*)} = \begin{bmatrix} 2\rho_C C_0^* - \tau_C - 2\alpha C_0^* & \rho_C C_0^* \\ 0 & \rho_L - \alpha C_0^* \end{bmatrix}, \tag{A5}$$

and as (A5) is an upper triangular matrix, eigenvalues are given by each element of the main diagonal, see Section 6.4 in [42]. Thus, by substituting $C_0^*$ and performing the corresponding arithmetic operations, results are indicated below:

$$\begin{aligned} \lambda_1 &= -\sqrt{\tau_C^2 + 4\varphi_C(\alpha - \rho_C)}, \\ \lambda_2 &= \alpha\tau_C + 2\alpha\rho_L - 2\rho_C\rho_L - \alpha\sqrt{\tau_C^2 + 4\alpha\varphi_C - 4\rho_C\varphi_C}. \end{aligned}$$

Theorem 4.7 by Khalil in [27] at Section 4.3 allows us to conclude that (4) is locally asymptotically stable if $Re\lambda_i < 0$, $i = 1, 2$. Therefore, one can solve $\lambda_2 < 0$ for the immunotherapy parameter $[\varphi_C]$ as follows:

$$\varphi_C > \varphi_{\inf} = \frac{1}{4(\alpha - \rho_C)} \left( \frac{\alpha\tau_C + 2\alpha\rho_L - 2\rho_C\rho_L}{\alpha} \right)^2 - \frac{\tau_C^2}{4(\alpha - \rho_C)},$$

and a necessary condition for tumor persistence (see Liu and Freedman in [55] at Section 3.1.3) is given by

$$\varphi_C < \varphi_{\inf}, \tag{A6}$$

and we formulate the next statement.

**Theorem A2.** *Persistence. If condition (A6) on the CAR-T cells therapy dose holds, then the ALL cancer cell population described by the system (1) and (2) persists.*

The latter implies that all solutions $L(t) > 0$ for all $t \geq 0$ as the CAR-T cells therapy dose will not be sufficient to completely eradicate the leukaemia cancer cells population. When substituting the parameter values from the scenario defined in Section 5 one gets the following value

$$\varphi_{\inf} = 25,866 \text{ CAR-T cells,}$$

which is lower than the therapy dose that was established by means of condition (6) to ensure the complete eradication of the tumour population, i.e., $\varphi_C > \varphi_{CART} = 28,187$ CAR-T cells, see Corollary 1 and Theorem 5.

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
