# Peer review of "CAR-T Cell Therapy for the Treatment of ALL: Eradication Conditions and In Silico Experimentation"

_hemato, doi:10.3390/hemato2030028_

Round 1

Reviewer 1 Report

This manuscript prepared well, and also the study is good and creative, it can be accepted after minor spell check.

Minor comments: suggeste to update the references and consider cite some lastest references in the suitable place, such as Liu et al. (2021). Algae-derived polysaccharides supplementation ameliorates heat stress-induced impairment of bursa of Fabricius via modulating NF-κB signaling pathway in broilers, Poultry Science, 100:101139, doi: 10.1016/j.psj.2021.101139; Zhao et al. (2021) Dietary Enteromorpha Polysaccharides Supplementation Improves Breast Muscle Yield and Is Associated With Modification of mRNA Transcriptome in Broiler Chickens. Front. Vet. Sci. 8:663988. doi: 10.3389/fvets.2021.663988.

Reviewer 2 Report

Brief summary:

In this article, the authors use theorems of the dynamic nonlinear system to investigate the behaviors of the interaction between acute lymphocytic leukemia (ALL) cancer cells and CAR-T cells, which is modeled as a two-species ODE system. Analytical derivation results in the lower and upper bounds of CAR-T cells and ALL cancer cells populations when the system achieves compact invariant sets, as well as the parameter conditions for the system to achieve the status. Additionally, sufficient conditions of parameter for the system to achieve the ensured eradication of ALL cancer cells. The analytical results are then analyzed with in silico experiments in the manuscript, and compared with clinical trials. Furthermore, alternative administration schedules of CAR-T therapy are numerically simulated and compared. This presented approach does represent an interesting framework to provide quantitative understanding of the dynamics of CAR-T cell therapy mechanism, and to provide information for the design of administration protocols of CAR-T cell therapy.    

Broad comments:

As the main strength of this paper, important theoretical conclusions are drawn regarding the relationship between the required dose and duration of administered CAR-T cell therapy to achieve complete eradication of ALL cancer cells, and the biological parameters such as killing efficacy rate, mitosis stimulation rate, natural death rate of CAR-T cells, and the proliferation rate of ALL cancer cells.

However, this study is mostly theoretical. Indeed, the value and range of parameters of interest are estimated by matching the numerical simulations of the required dose and complete remission period with related reports from a previous dose-escalation trial, and then compared with literature values. But the actual application and validation of this theory requires accurate measurement of involved biological parameters, which is difficult to achieve in real animal experiments or clinical practice, and not well addressed. The discussion on applying this approach in the design of personalized strategies is not strong enough. To claim this conclusion drawn in the manuscript, the authors should at least discuss the possible approaches to acquire data for individual patients required for parameter calibration, as well as prediction validation, or perform uncertainty quantification regarding these parameters.    

Specific comments:

  1. Introduction (lines 60 - 62): The last sentence of this paragraph would benefit from citing previous experimental studies or clinical trials indicating the effects of administration doses and schedules of CAR-T cell therapy on the toxicity and treatment efficacy.
  2. Materials and methods (lines 123 – 124): Typo in the sentence, “L_fh(x) represents the Lie derivative (temporal derivative) of f(x)”. Isn’t it supposed to be the temporal derivative of h(x)?
  3. The ALL and CAR-T cells mathematical model (lines 214): The justification of value and range of involved parameters should be mentioned in the content where Table 1 is first introduced, or explicitly pointed out that this will be addressed in later sections.
  4. Discussion and in silico experimentation (lines 261 – 265): Did the authors perform sensitivity analysis to evaluate how the variation of these parameters would affect either analytical solutions or numerical simulation results? If so, please briefly describe it in the Results section or include it in the Appendix.
  5. Conclusion (line 379): The discussion to address this conclusion is not strong enough. To claim the potential to apply this approach in the design of personalized strategies, the authors should at least discuss the possible approaches to acquire data for individual patients required for parameter calibration, as well as prediction validation, or perform uncertainty quantification regarding these parameters.  

Reviewer 3 Report

Valle et al. present mathematical modeling systems to predict best outcome for CART therapy.

Minor comments:

 Page 1 ll30 rather elderly in general (patients >74 years have also very poor outcome.

Page 1 ll 31/32 should be deleted as it adds nothing to the manuscript.

Treatment strategies should be more detailed including bispecific antibodies and conjugated antibodies.

Overall the paper is very interesting and original.
